# A Comparison of Veterans with Problematic Opioid Use Identified through Natural Language Processing of Clinical Notes versus Using Diagnostic Codes

**DOI:** 10.3390/healthcare12070799

**Published:** 2024-04-06

**Authors:** Terri Elizabeth Workman, Joel Kupersmith, Phillip Ma, Christopher Spevak, Friedhelm Sandbrink, Yan Cheng, Qing Zeng-Treitler

**Affiliations:** 1Washington DC VA Medical Center, Washington, DC 20422, USA; 2Biomedical Informatics Center, The George Washington University, Washington, DC 20037, USA; 3School of Medicine, Georgetown University, Washington, DC 20007, USA

**Keywords:** opioid misuse documentation, Veterans, natural language processing, machine learning, demographic data, comorbidities, care use

## Abstract

Opioid use disorder is known to be under-coded as a diagnosis, yet problematic opioid use can be documented in clinical notes, which are included in electronic health records. We sought to identify problematic opioid use from a full range of clinical notes and compare the demographic and clinical characteristics of patients identified as having problematic opioid use exclusively in clinical notes to patients documented through ICD opioid use disorder diagnostic codes. We developed and applied a natural language processing (NLP) tool that combines rule-based pattern analysis and a trained support vector machine to the clinical notes of a patient cohort (n = 222,371) from two Veteran Affairs service regions to identify patients with problematic opioid use. We also used a set of ICD diagnostic codes to identify patients with opioid use disorder from the same cohort. The NLP tool achieved 96.6% specificity, 90.4% precision/PPV, 88.4% sensitivity/recall, and 94.4% accuracy on unseen test data. NLP exclusively identified 57,331 patients; 6997 patients had positive ICD code identifications. Patients exclusively identified through NLP were more likely to be women. Those identified through ICD codes were more likely to be male, younger, have concurrent benzodiazepine prescriptions, more comorbidities, and more care encounters, and were less likely to be married. Patients in both these groups had substantially elevated comorbidity levels compared with patients not documented through either method as experiencing problematic opioid use. Clinicians may be reluctant to code for opioid use disorder. It is therefore incumbent on the healthcare team to search for documentation of opioid concerns within clinical notes.

## 1. Introduction

Opioid misuse is a serious, escalating public health issue. Since 1999, there have been 932,000 overdose deaths in the United States, mainly driven by opioids [1]. This has been exacerbated by the COVID-19 pandemic; in 2020 preventable opioid overdose deaths increased 41% [2]. Veterans of the U.S. military may especially be harmed by the misuse of opioids. Veterans are more likely to experience opioid poisoning mortality [3] and face increasing rates of opioid use disorder (OUD) [4] and opioid overdose death [5].

To prevent overdose and other opioid complications, it is imperative to identify patients experiencing problematic opioid use. Electronic health records (EHRs) can serve as a data source, and include structured data (e.g., ICD diagnostic codes) and unstructured data, as found in free text clinical notes. Relying only on structured data is insufficient because opioid overdose is under-coded [6], and structured data are insufficient for identifying OUD [7]. Identifying patients experiencing problematic opioid use through EHR data could provide early detection of cases that are otherwise currently missed.

Natural language processing (NLP) can extract information from free text and generally consists of trained machine learning models, hard-coded rules matching patterns in text, or both. NLP has been applied to clinical notes to identify documentation of opioid use issues, usually for specific clinical settings or patient types. Blackley et al. randomly selected emergency department notes, inpatient progress notes, and previous hospital discharge summaries to develop a rule-based model plus several machine learning models to classify inpatient encounters by OUD status, with best performance achieved by a random forest classifier yielding 97% accuracy [8]. Using the EHR data and notes of chronic opioid therapy patients, Carrell et al. developed an NLP application to identify OUD patients not otherwise identified through ICD codes, with a focus on maximizing sensitivity, yielding a 34.6% false positive rate for documents and a 41% false positive rate for patients [9].

In other studies, Afshar et al. used inpatient notes for patients screened for illicit drug use to develop a convolutional neural network classifier, achieving 81% sensitivity and 72% positive predictive value (PPV) [10]. Zhu et al. developed a rule-based system achieving 98.5% precision, 100% recall, and 99.2% F1, but also limited their study to the notes of chronic opioid therapy users [11]. In research pursuant to data de-identification, Sharma et al. [12] developed diverse classifiers using inpatient encounters, experimenting with several different types of features. For models having n-gram features, a convolutional neural network achieved an F1 score of 84%, 94% PPV, 75% recall, 98% specificity, and 88% negative predictive value. An n-gram is a word sequence in text, where the n references the word count. For example, “prescription” is a unigram, and “primary provider” is a bigram. Poulsen et al. developed logistic regression and deep learning tools to identify and characterize OUD in hospital discharge summaries from the MIMIC III database [13]. Performance ranged according to class, with the deep learning tool’s performance achieving the best performance ranging in F1 scores from 48 to 99. Kashyap et al. also used MIMIC III database notes associated with intensive care unit admissions [14] to develop logistic regression and deep learning models to identify opioid prescription likelihood and OUD in clinical notes. On the OUD identification task, their deep learning model achieved an F1 score of 0.82 +/− 0.05 and AUCROC of 0.94 +/− 0.008. Gabriel et al. [15] developed an NLP multilayer perceptron tool that leveraged ClinicalBert to predict persistent opioid use [16]. Using outpatient surgical notes, their tool achieved an F1 score of 0.909 and AUC of 0.839 in validation testing.

Many of these prior studies targeted patients who already had a recognizable risk for problematic opioid use, which does not take full advantage of information for all patients and all clinical settings available in EHR data sources. This begs the question of whether the characteristics of patients who received relevant diagnostic codes are different from all patients, from all clinical settings, identified as at risk for problematic opioid use using clinical notes. To answer this, using clinical notes from all encounter types, and all types of patients would provide an expanded view of problematic opioid use documentation in EHR data.

The objectives of this study were to identify problematic opioid use from clinical notes from all available clinical settings and patients and to analyze the differences between patients identified exclusively through NLP classification, and patients identified using ICD codes, regarding demographics, selected comorbidities, prescription data, and healthcare encounters. Our initial hypothesis was that there would be a difference between these two groups. To pursue this, we developed an open-source NLP tool (https://github.com/GWU-BMI/opioids-nlp) (accessed 5 April 2024) to identify evidence of problematic opioid use in clinical notes from a broad range of clinical settings and patients. We also analyzed structured data to identify patients receiving an OUD ICD code. Patient EHR records from the U.S. Department of Veterans Affairs (VA), stored within the VA central data warehouse (CDW), served as the data source. We accessed these data, with IRB approval, for two VA service regions within the secure VINCI research platform [17].

## 2. Materials and Methods

### 2.1. Study Cohort

We identified a cohort of 222,371 patients with at least 2 outpatient encounters within the Baltimore, Maryland VA station or the Washington, DC VA station between 1 January 2012 and 31 December 2019. A VA station is a regional service area that typically includes at least one hospital and additional resources such as outpatient clinics. These patients had a total of 81,129,781 notes from all clinical settings over that period.

### 2.2. Annotation Guideline

To better understand problematic opioid use documentation in clinical notes, two members of the research team independently annotated 196 snippets containing 46 relevant keywords or key phrases (collectively noted as “key phrases”), including generic and trade names for opioid drugs, and phrases like “tapered”, “withdrawal”, and “opioid abuse” for positive problematic opioid use, suspected problematic opioid use, and negative documentation. This initial list was assembled using findings from previous work and professional clinical knowledge [18,19]. A positive classification arose from documentation suggesting current abuse, overuse, or addiction to prescribed or illicit opioids. This positive documentation was generally conveyed through readily recognized clinical or standardized language, such as “overuse of opioid medication” or “admitted to abuse of prescription pain killers in particular oxycodone” or “6. opioid abuse (snomed ct 5602001)” in an active problems list, or a string of text like “polysubstance dependence (opioid, cannabis and cocaine)…uses…lorcet (600–1000 mgs) daily” or “current use:..heroin (intranasal), roxicet”. Negative documentation was where a key phrase was present, but there was no indication of problematic opioid use in the text. Each snippet consisted of the key phrase and the 50 words before it and after it in the document. The snippets were extracted from randomly selected notes containing the key phrases. After calculating inter-rater agreement (86.73%), the two team members reviewed and discussed the results. They then independently reviewed 185 different snippets containing the key phrases, measured inter-rater agreement (97.31%), and again discussed the results. They then identified the most relevant key phrases for problematic opioid use documentation, based on their findings.

### 2.3. NLP Tool Development

The research team developed an NLP tool that combines supervised machine learning and rule-based pattern recognition to classify snippets. The machine learning component utilizes the position of the key phrase in the text (in cases where there are multiple words in the key phrase, the position of the first word in the key phrase is used) and n-grams as features. The text of all snippets serving as input to the machine learning component was preprocessed by transforming it to lower case, then it was tokenized, and non-alphanumeric characters were removed. Using 582 annotated snippets classified by the research team, a support vector machine (SVM) trained using as features 946 unigrams that consisted entirely of letters and 474 bigrams where the first word consisted of letters and the second word consisted of letters, numbers, or a combination of letters and numbers yielded the best initial performance out of several different algorithms tried (SVM, Random Forest, AdaBoost). All experiments utilized the Python scikit-learn package, version 1.0.2. To assess performance, we measured precision/PPV and sensitivity/recall using an 80%/20% train/test split of the data. Splitting the data in this fashion enables testing using “unseen” data, i.e., data not used to train a given model. The best SVM achieved 94% precision/PPV and 84% sensitivity/recall, whereas the best random forest model achieved 89% precision/PPV and 78% recall/sensitivity and the best Adaboost achieved 85% precision/PPV and 80% sensitivity/recall. Snippets annotated as having suspected problematic opioid use were aggregated with snippets annotated as having positive problematic opioid use to serve as positive training examples; snippets annotated as negative documentation of problematic opioid use were used as negative training examples.

To automatically identify template data and standard language found in opioid documentation, we also developed a library of 145 regular expressions. A regular expression is a string of specialized characters forming a pattern to match specific types of text. For example, a regular expression like ‘\[\s*x\s*\]’ could be used to match a questionnaire element like “[x]” and account for potential white space around “x”. Examples of noteworthy content that were identified using regular expressions include statements like “former opioid dependence”, “sister abuses hydrocodone”, and template data like “[x] substance abuse and/or dependence”. In these examples, the phrase “former opioid dependence” indicates opioids are a former, i.e., not a current issue for the patient, and “sister abuses hydrocodone” references someone other than the patient. The phrase “[x] substance abuse and/or dependence” is from a questionnaire that directly pertains to the patient at the time it was recorded. To fine-tune the library of expressions, 160 additional documents containing the final key phrases—40 each from the even years of the study—were randomly selected and reviewed for additional relevant patterns.

Rule-based pattern recognition and machine learning classification provided the operative power of the NLP tool. It received snippets containing the key phrases as input. It first analyzed their content for patterns recognized by the library of regular expressions in the rule-based classifier. This module used a sequential voting system to classify snippets containing these patterns. Regular expressions were grouped by what they identified: *absolute positive patterns*, *canceling patterns*, *general positive patterns*, and *neutral patterns*. These patterns were applied in a sequential manner: First, snippets containing an absolute positive expression like “patient continues to struggle with opioid abuse” (i.e., indicating the patient has current problematic opioid use issues) were matched by an *absolute positive pattern*, classified as positive, and transferred to output. Second, the remaining snippets containing any canceling expressions like “former opioid dependence” (i.e., not current) were then matched by a *canceling pattern*, classified as negative, and transferred to output. Third, the remaining snippets containing any general positive expressions like “opioid abuse” were then matched by a *general positive pattern*, classified as positive, and transferred to output. Finally, the remaining snippets containing neutral patterns like “sister abuses hydrocodone” (i.e., someone other than the patient) were then matched by a *neutral pattern*, classified as negative, and transferred to output. Figure 1 illustrates this process.

After the rule-based classifier had classified all snippets containing text matching the regular expression patterns, the trained SVM model classified the remaining snippets after the preprocessing described earlier. Figure 2 illustrates the NLP tool’s overall procedure for classifying snippets.

### 2.4. Evaluation

A different, unseen test set of 161 annotated snippets not used in the NLP tool’s development was used to evaluate the classifier. These snippets consisted of randomly selected data containing one of the key phrases, including randomly selected data containing template markup from the odd years of the study. By including template data from odd years in the testing, we also included potentially new template patterns for the testing process. None of the test-set snippets were used to develop or train the model. We measured recall/sensitivity, specificity, precision/PPV, and overall accuracy.

### 2.5. Classifying Cohort Notes

We established a minimum performance threshold of 85% for each of the four metrics, which has been recognized as a useful performance threshold in machine learning [20]. If this minimum performance level was achieved by the NLP tool in classifying the unseen test set, we would then apply it to all clinical notes from the cohort.

### 2.6. Statistical Analysis

For each of the patient groups, we retrieved demographic data, selected comorbidity and prescription data, and outpatient visit counts. The main outcome was problematic opioid use, and how it had been identified. We conducted a two-way *t* test to compare average ages at the index date and average outpatient encounter visits after the index date, and Chi-square tests for the baseline comorbidities, prescription data, and the rest of the demographic data. The index date for each patient was the first date of documentation of problematic opioid use, either through ICD code or the NLP tool. These analyses were carried out using SAS, version 9.4 (Cary, NC, USA). Because of the large sample size (n = 222,371), a small difference between groups may be statistically significant. Therefore, we also calculated the absolute standardized difference (ASD) for all variables. An ASD > 10% indicated significant imbalanced characteristics between the two groups [21,22].

### 2.7. Grouping Patients by Identification Method

We used the classifier output to identify patients having problematic opioid use as indicated by NLP. We identified patients diagnosed with OUD as indicated by ICD codes (ICD-CM-9: 304.00, 304.70. 305.50; ICD-CM-10: F11), using the results of the structured data analysis. Patients identified as having problematic opioid use only by the classifier (referred to as *NLP Only*) constituted one group for analysis. Patients identified as having problematic opioid use as indicated by ICD code (referred to as *All ICD*) constituted the other group for analysis. Therefore, by using this strategy, some of the All ICD group patients may also have had one or more snippets identified as positive by the classifier, but none of the NLP Only patients received a relevant ICD code in the study period, and both groups were mutually exclusive. For comparative purposes, we also identified cohort members not identified as having problematic opioid use by either ICD code or NLP (referred to as *No Problematic Opioid Use*).

### 2.8. Prominent Note Types among Patient Groups

As an additional measure to shed light on differences between patients identified as having problematic opioid use only through NLP (NLP Only) and those who were identified through relevant ICD codes that also had clinical notes positive for problematic opioid use (referred to as *NLP/ICD*), all clinical notes identified via NLP as positive for problematic opioid use for patients in both groups were retrieved. We analyzed multiple properties of the notes and their snippets.

## 3. Results

### 3.1. Key Phrases

In building the annotation guidelines, 36 key phrases relevant to problematic opioid use were identified. Table 1 contains these key phrases.

### 3.2. Classifier Performance

The classifier achieved 96.6% specificity, 90.4% precision/PPV, 88.4% sensitivity/recall, and 94.4% accuracy. This performance satisfied the threshold we had previously established. We also felt it was sufficiently comparable to that of other studies, considering the expanded task of accommodating data from all clinical settings and patients.

### 3.3. Clinical Note Classification

The NLP tool was used to classify 3,521,637 notes of cohort patients (notes containing one or more of the key phrases). These notes contained 8,804,031 snippets, each including one of the key phrases. Table 2 indicates these results on a document basis. Table A1 in the Appendix A includes counts for each key phrase in the snippets. Table A2 in the Appendix A includes the number of patients having an occurrence of each key phrase in a snippet.

Table 3 contains example text spans from positive and negative snippets and the classification method used. Key phrases are highlighted in yellow. The positive examples are indicative of the types of positive documentation sought according to the study’s goals.

The positive examples identified through regular expressions include “opioid dependence” in an enumerated problem list, “opioid dependence” included as a current diagnosis, plus a snippet that included both opioid dependence in the patient’s current problem list and an allergy to Demerol. In this last example, the “opioid dependence (icd-9-304.00)” was identified as an “absolute positive expression” first, so the snippet was classified as positive, despite also documenting an allergy to an opioid drug, which constituted a “neutral expression”. The positive examples identified through the SVM include “substance abuse”, “oxycodone” (and “morphine”), and “withdrawal”. The three negative examples identified through regular expressions include directions for taking medication, a family history of substance abuse (that did not pertain directly to the patient), and an opioid medication included in a list of current medications. The negative machine learning examples include a patient’s experience of taking the drug where she developed a rash, a patient that did not request the drug, and directions for taking medications (including oxycodone).

### 3.4. Problematic Opioid Use in Patients

The NLP tool identified 63,574 patients that received 1 or more positive snippets relating to problematic opioid use. Of all patients positively classified through NLP, 57,331 received a positive classification exclusively by that method (i.e., NLP Only). Within the cohort, 6997 patients had received 1 or more ICD diagnostic codes for OUD (i.e., All ICD) (6243 patients in the All-ICD group also had one or more positive snippets). There were differences between the NLP Only and All ICD groups regarding demographic attributes (Table 4), comorbidities, prescription data, and outpatient visits (Table 5). In these tables, ASD > 10% (bold text) indicates significant imbalanced characteristics between the two groups. For comparison, the values of cohort members not classified by NLP or ICD code (i.e., No Problematic Opioid Use) are included, along with *p*-values and ASD compared with the NLP Only group in Table 4 and Table 5.

Table 4 displays several differences among the NLP Only and All ICD groups. Patients in the All ICD group were less likely to be married and more likely to be divorced or separated than the NLP Only group. Patients in the All ICD group were more likely to be male and younger. Patients in the NLP Only group were more likely to be female. Other significant variable attributes, according to ASD, include race and ethnicity.

Comorbidity data (Table 5) reveal notable trends, where, with the exception of cancer, hypertension, and diabetes, the All ICD group had significantly higher comorbidity levels compared with the NLP Only group, according to ASD. This same trend was evident for prior opioid prescriptions, concurrent benzodiazepine prescriptions, and outpatient visits during the study period. All values in Table 5 reveal a continuum of escalating comorbidity levels, prior opioid and current benzodiazepine prescriptions, and care encounters according to group type.

### 3.5. NLP Classifications among Notes

Table 6 includes patient counts on positive snippet classifications and mean positive notes for the NLP Only patients and the patients identified through ICD code that also had clinical notes positive for problematic opioid use (i.e., NLP/ICD, n = 6243). For the NLP Only group, a positive classification was more likely to be incurred by a note containing a key phrase like ‘opioid abuse’ or ‘withdrawal’ than a note containing a specific drug name like ‘hydrocodone’. However, the NLP/ICD group document counts in these categories were more similar. The mean positive snippet count for the NLP/ICD group was almost twice that of the NLP Only group.

### 3.6. Predominant Note Types

There was also variation in the frequent positive clinical note types for the NLP Only patients, and the patients identified through ICD code that also had clinical notes positive for problematic opioid use (NLP/ICD). Figure 3 and Figure 4 include bar charts of the 25 most frequent note types for each group.

The most frequent documents for the NLP Only group (Figure 3) address unique issues like opioid prescribing and homelessness. Both charts include note types for substance abuse treatment (i.e., SATP, substance abuse treatment program; SARP, substance abuse rehabilitation program), but there are several in the NLP/ICD group (Figure 4). Both charts include several mental health-oriented notes, emergency department notes, and notes for other health issues (diabetes in the NLP Only group; eye clinic in the NLP/ICD group).

## 4. Discussion

This research produced an effective method to identify problematic opioid use among all patients. The findings in demographic data, comorbidities, prescription data, and care usage, as well as the note trends, confirmed there were differences between patients receiving an OUD ICD code (All ICD group) and those having problematic opioid use documented only in clinical notes (NLP Only group). By patient count, these findings also suggest that problematic opioid use is more likely to be documented in clinical notes than through ICD diagnostic codes.

The results indicate differences between the NLP Only and All ICD patient groups in regard to demographics (Table 4). Veterans in the All ICD group were significantly younger, and both groups were younger than the mean age of 58.3 of patients having no positive NLP or ICD classification. This finding correlates with previous research [23,24]. Veterans within the NLP Only group were more likely to be female. Other research has found gender differences in opioid misuse communications, with women more likely than men to report opioid issues [25]. This difference may also be reflected in what providers document in clinical notes. Women are also more likely to receive opioid prescriptions on outpatient visits than men [26] and are more likely to be prescribed opioids for chronic conditions [27]. The patients in the All ICD group were less likely to be married. Chronic pain sufferers who are not married are more likely to abuse opioids [28], possibly due to less social support. The issue of race in opioid misuse and abuse must be examined through a contextual lens. Issues such as bias [29], and inequities of treatment and resources [30,31,32] must be considered in this larger societal discussion. In this study, race was a significant factor; this should also be interpreted within the framework of potential bias and access inequities. All of the significant demographic findings warrant future study.

There were notable differences in comorbidities between the All ICD and NLP Only groups (Table 5). This analysis includes a variety of comorbidities, including pain-oriented conditions, mental health issues, addictions to other substances, and chronic diseases. The correlation between pain and opioid use issues had been studied [28], as well as that between opioid abuse and misuse of other substances [33]. This current study highlights the differences between prevalence of these comorbidities in terms of the methods clinicians use to document problematic opioid use. Patients in the All ICD group were also more likely to have concurrent benzodiazepine prescriptions, to have prior opioid prescriptions, and to receive more overall outpatient treatment. Levels in the All ICD and NLP Only groups exceeded those in the No Problematic Opioid Use group. Collectively, the data suggest that the NLP Only group is worse off than patients having no documented problematic opioid use, and is approaching the state of the All ICD group. This is especially concerning, since the NLP Only patients have no recorded ICD code to enable providers to recognize their opioid-related issues.

The findings in Table 6 and Figure 3 and Figure 4 suggest different clinical note documentation patterns. For patients without a recorded OUD ICD diagnostic code, providers tended to discuss problematic opioid use outside of the context of specific drugs. For those with an OUD ICD diagnostic code, providers documented problematic opioid use in the context of specific drug names almost as often as by using other types of key phrases. Providers were also more likely to document problematic opioid use in clinical notes once an OUD ICD code had been recorded. It is possible that providers felt more comfortable recording problematic opioid use once a structured data element was on record. Note types also varied by group. In the NLP Only group, notes addressing opioid prescribing and homelessness were prominent, whereas notes from substance abuse treatment groups were more prominent in the NLP/ICD group. This last observation suggests that patients receiving an ICD code are more likely to receive treatment.

These findings also suggest subgroups among Veterans for increased awareness of potential opioid misuse concern. These subgroups include women, people who are not married, and people of color. As a part of this awareness, there should also be cognizance of potential bias and the inequities of available OUD resources for the patient, and a plan to address this.

This work addresses a gap in opioids research, namely, differences between patients according to problematic opioid use documentation methods utilizing all types of available clinical notes. Singleton and colleagues [34] recently addressed this issue of documentation method, creating a rule-based NLP tool that achieved 81.8% sensitivity and 97.5% specificity when applied to ED triage, ED general, history and physical, addiction medicine consultation, and discharge summary notes. Their study, which was visit-based, also found overlap among NLP vs. ICD classification, with some being classified only through NLP. This study expands on this notion of documentation method by analyzing all available note types and examining additional factors like patient comorbidities and prescription data.

The NLP tool achieved satisfactory results (96.6% specificity, 90.4% precision/PPV, 88.4% sensitivity/recall, 94.4% accuracy) according to our original minimum performance threshold. It performed as well as many newer and possibly more sophisticated machine learning methods, which is especially significant considering that, unlike the other studies cited, we developed it for and applied it to all available note and patient types.

Finally, our findings suggest that clinicians may be reluctant to code for opioid use disorder due to negative implications perceived by both clinicians and patients, as evidenced by the patient counts in the NLP Only and All ICD groups. The study took place in the greater Washington DC area where there are large percentages of Veterans and civilians having formal associations with government, including employment. There may be a perception that having a diagnosis of OUD in the medical record may jeopardize these relationships. It is therefore incumbent on the healthcare team to search for documentation of opioid use within the clinical notes.

### 4.1. Limitations

The NLP tool performance was tested on a small set of VA clinical notes in comparison to the amount of notes it was used to classify. However, in reviewing the classification results, it was evident that the tool performed well, enabling the realization of the study’s objectives and completion of the study. The generalizability to other notes from other clinical settings will be tested in the next step of our research. Overall, VA patients are skewed in age, gender, and possibly race and other characteristics. This also may affect the generalizability of the findings; however, the younger Veteran population (opioid users and abusers tend to be younger [35]) is more like that of the general U.S. population.

### 4.2. Future Work

This work is part of a larger project addressing concordant care [36]. As part of a larger project, we will test the NLP tool which was developed using VA data with MedStar Health System data. MedStar Health is a large healthcare system in the Mid-Atlantic region, with 491 facilities, including 10 hospitals and many clinics. We plan to further refine the NLP tool as needed. The NLP tool from this study has been made available as open-source software. Additional revisions will be posted in the future. Future work will also include explorations of other NLP methods, including experimentations with other machine learning approaches including deep learning, and possibly large language models. We also hope to explore additional variables, namely social determinants of health, and problematic opioid use documentation in future work. This will include development of an improved NLP tool to identify this information in clinical notes.

## 5. Conclusions

We developed an efficient NLP tool to detect problematic opioid use and applied it to clinical notes in order to better understand differences between patients with relevant unstructured documentation, compared to those with a structured OUD diagnosis, in EHR records. We found significant differences in terms of demographic and comorbidity characteristics, prescriptions, outpatient visits, and documentation patterns. The findings suggest differences by medium—i.e., structured data or unstructured clinical notes—in how clinicians document problematic opioid use in patients.

## Figures and Tables

**Figure 1 healthcare-12-00799-f001:**
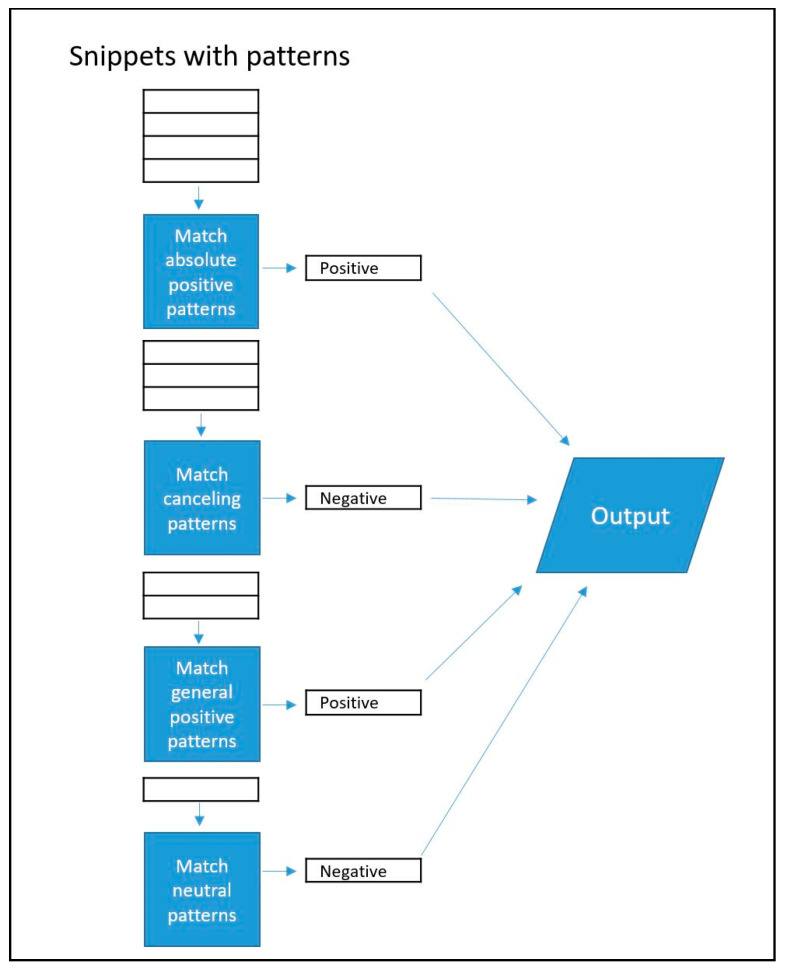
Rule-based classification. Beginning at the top, snippets with relevant patterns are classified sequentially according to pattern type.

**Figure 2 healthcare-12-00799-f002:**
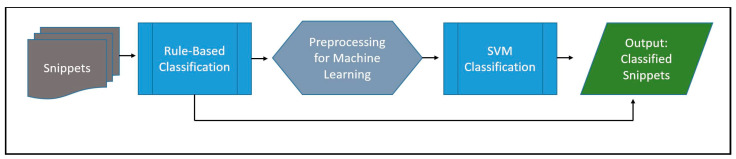
NLP tool pipeline operations. Raw snippets are first processed by the rule-based module, which sends the snippets it classifies to output. Snippets that are not classified by the rule-based module are preprocessed and used as input for the trained SVM model, which trains all the remaining snippets and sends them to output.

**Figure 3 healthcare-12-00799-f003:**
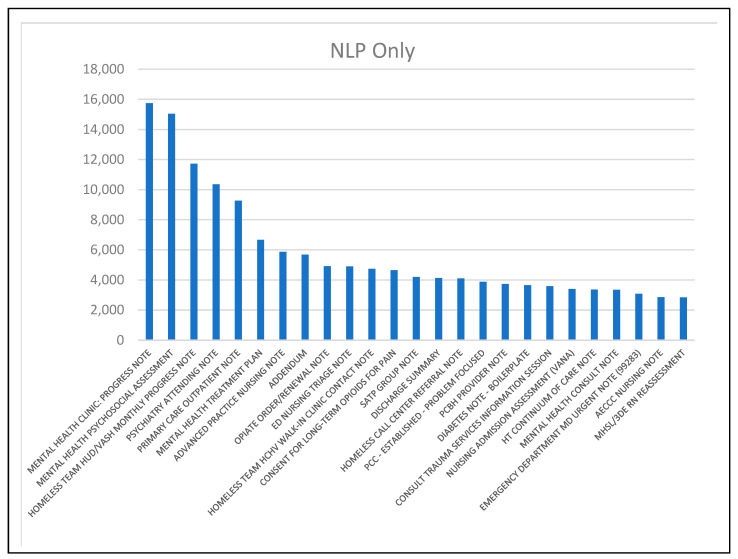
Prominent positive clinical note types for NLP Only group.

**Figure 4 healthcare-12-00799-f004:**
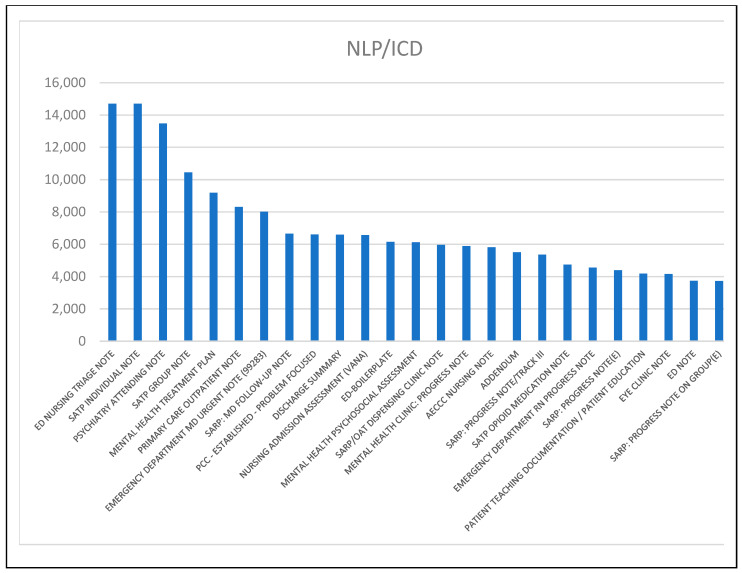
Prominent positive note types for patients receiving an ICD code who also had positive clinical notes.

**Table 1 healthcare-12-00799-t001:** Key phrases.

Key Phrases					
abstral	duragesic	hysingla	methadose	oxaydo	withdrawal
actiq	exalgo	kadian	morphine	oxycodone	zohydro
demerol	fentanyl	lorcet	norco	oxycontin	opioid dependence
dependence	fentora	lortab	opiate	percocet	polysubstance abuse
dilaudid	hydrocodone	meperidine	opiate abuse	roxicet	substance abuse
dolophine	hydromorphone	methadone	opioid	vicodin	substance dependence

**Table 2 healthcare-12-00799-t002:** Clinical note classification, document basis.

Element	Total
Years (2012–2019)	8
Key phrases	36
Total notes	3,521,637
Total snippets	8,804,031
Positive snippets	1,885,642
Negative snippets	6,918,389
Mean snippets per document	2.9

**Table 3 healthcare-12-00799-t003:** Positive and negative snippet text examples, with classification method.

Positive for Problematic Opioid Use and Classification Method		Negative for Problematic Opioid Use and Classification Method	
…substance abuse treatment…heroin last used: “yesterday”…	Machine learning	…pt has pain mostly at night was on Lorcet and tried to change to morphine but since she developed rash…	Machine learning
…4. low back pain…5. opioid dependence…6. homeless single person…	Regular expression	...hydromorphone 4 mg tab take one tablet every four active hours when needed for pain…	Regular expression
…opioid dependence (icd-9-cm 304.00)…	Regular expression	…family hx of substance abuse…	Regular expression
Alludes to the possibility of self medicating on the street…opiate withdrawal	Machine learning	…patient requested no Lortab…	Machine learning
…would not receive prescription for morphine and oxycodone until next month…reiterated multiple times that taking additional doses of opiates was a patient safety issue and would not be tolerated…	Machine learning	…continue Tylenol and oxycodone as needed per home regimen…	Machine learning
...allergies: darvon, periactin, phenothiazine/related antipsychotics, demerol…opioid dependence (icd-9-cm 304.00)	Regular expression	…9) hydromorphone inj, soln active…give: 0.5 mg/0.5 mL ivp q2h prn…for pain…	Regular expression

**Table 4 healthcare-12-00799-t004:** Demographic Data. Significant ASD values are in bold text.

	All ICD	NLP Only	*p*-Value (All ICD vs. NLP Only)	ASD (All ICD vs. NLP Only)	No Problematic Opioid Use	*p*-Value (NLP Only vs. No Problematic Opioid Use)	ASD (NLP Only vs. No Problematic Opioid Use)
N	6997	57,331			158,043		
Gender%			<0.0001			<0.0001	
M	93%	82%		**34**	84.9%		8
F	7%	18%		**34**	15.1%		8
Mean Age/Standard deviation (at year patient entered cohort)	53.3/ 12.2	55.4/ 16.1	<0.0001	**15**	58.8/18.7	<0.0001	**17**
Marital Status%			<0.0001			<0.0001	
Married	25.7%	38.5%		**28**	50.2%		**24**
Divorced	31.6%	25.8%		**13**	17.1%		**21**
Never Married/Single	26.5%	22.8%		9	15.6%		**18**
Widowed	4.5%	5.1%		3	6.9%		8
Separated	11.3%	6.5%		**17**	3.2%		**16**
Missing/Other	<1.0%	1.3%		9	6.9%		**29**
Race%			<0.0001			<0.0001	
Black/African American	59.7%	54%		**11**	28.2%		**54**
White	35.7%	36.6%		2	51.4%		**30**
Asian	0.1%	1.0%		**12**	1.2%		2
Native Hawaiian/Pac. Islander	<1.0%	<1.0%		1	<1.0%		2
American Indian/Alaska Native	<1.0%	<1.0%		3	<1.0%		1
Unknown	3.6%	7.2%		**16**	18.2%		**34**
Ethnicity%			<0.0001			<0.0001	
Not Hispanic or Latino	96.5%	92.3%		**19**	80.6%		**35**
Hispanic or Latino	1.5%	2.9%		9	2.9%		<1
Unknown	1.9%	4.9%		**16**	16.5%		**38**

**Table 5 healthcare-12-00799-t005:** Comorbidity, opioid prescription history, concurrent benzodiazepine prescription data, and means and standard deviations of outpatient encounters. Significant ASD values are in bold text.

	All ICD	NLP Only	*p*-Value (NLP Only vs. All ICD)	ASD (%) (NLP Only vs. All ICD)	No Problematic Opioid Use (%)	*p*-Value (NLP Only vs. No Problematic Opioid Use)	ASD (NLP Only vs. No Problematic Opioid Use)
N	6997	57,331			158,043		
Comorbidities (when or after patient entered cohort)							
Hypertension	57.1%	53.5%	<0.0001	7	45.8%	<0.0001	**15**
Diabetes mellitus	22.1%	25.0%	<0.0001	7	20.4%	<0.0001	**11**
Depression	61.8%	41.1%	<0.0001	**42**	19.6%	<0.0001	**48**
Post-traumatic stress disorder	39.6%	25.1%	<0.0001	**31**	10.8%	<0.0001	**38**
Cancer	9.0%	12.1%	<0.0001	10	12.9%	<0.0001	2
Tobacco	62.0%	31.1%	<0.0001	**65**	15.4%	<0.0001	**38**
Alcohol	60.9%	23.8%	<0.0001	**81**	9.2%	<0.0001	**44**
Other drug addictions	66.6%	18.6%	<0.0001	**111**	4.2%	<0.0001	**47**
Traumatic brain injury	11.5%	7.0%	<0.0001	**16**	3.6%	<0.0001	**15**
Anxiety	39.7%	27.6%	<0.0001	**26**	14.1%	<0.0001	**34**
Neck pain	39.4%	31.3%	<0.0001	**17**	18.2%	<0.0001	**31**
Back pain	57.0%	47.2%	<0.0001	**20**	30.7%	<0.0001	**34**
Prior VA opioid prescription	71.5%	51.6%	<0.0001	**42**	32.1%	<0.0001	**40**
Concurrent benzodiazepine prescriptions	19.4%	10.1%	<0.0001	**26**	3.9%	<0.0001	**24**
Mean/standard deviation outpatient encounters (since patient entered cohort; max 1 per day)	50.9/ 49.8	33.4/ 31.3	<0.0001	**42**	16.2/23.2		**62**

**Table 6 healthcare-12-00799-t006:** Patient counts by note key phrase type for patients with positive NLP notes; positive note count mean and standard deviation for patients with positive NLP notes.

NLP Only, NLP/ICD Patient Groups, Positive Snippet Classifications		
	Count of patients having positive snippets with specific drug name	Count of patients having positive snippets with other key phrases
NLP Only	15,495	54,856
NLP/ICD	5298	6175
	Positive Snippets Mean/Standard Deviation	
NLP Only	1.7/1.5	
NLP/ICD	3.0/2.9	

## Data Availability

The data used in this study are not publicly available because they include protected health information.

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
