# Peer review of "A Comparison of Veterans with Problematic Opioid Use Identified through Natural Language Processing of Clinical Notes versus Using Diagnostic Codes"

_healthcare, 2024, doi:10.3390/healthcare12070799_

Round 1

Reviewer 1 Report

Comments and Suggestions for Authors

1- The abstract did not explain the working structure of the proposed system. In addition to not explaining the result of this system and the measure of its efficiency

2- It is better to choose keywords that are different from the words in the search title

3- The researcher's contributions are unclear. Please explain it more clearly and make it in point form

4-Where are the previous studies in this field? Please add it in the form of research to clarify the series of previous attempts before converting it to the form the researcher presents here

5- Where are the proposed work algorithms? It should be explained in the form of an algorithm or flowchart

6-The number of references is relatively small

7-The topic is modern in terms of its presentation. Still, it was not presented clearly, and the practical approach was not clarified in the accepted format when working on building an automated learning system. However, it requires a little organization and clarification for each detail of the work to be perfectly ready for publication. Likewise, the work did not address any previous works, but this matter is important because any work must be preceded by previous works that gave him an idea to build future work.

Author Response

We thank you for your thoughtful review of the manuscript.  Below are responses to your comments.

1- The abstract did not explain the working structure of the proposed system. In addition to not explaining the result of this system and the measure of its efficiency

  • We have added text to the abstract that further describes the NLP tool and its performance. We did this in a manner to not augment the abstract too far from the 200 word limit.

2- It is better to choose keywords that are different from the words in the search title

  • We added a few key phrases that are significant to the article but are not in the title.

3- The researcher's contributions are unclear. Please explain it more clearly and make it in point form.

  • The contributions follow the format prescribed by the journal in https://www.mdpi.com/journal/healthcare/instructions . We are not sure how to proceed, so we defer to the Section Manager Editor for changes.

4-Where are the previous studies in this field? Please add it in the form of research to clarify the series of previous attempts before converting it to the form the researcher presents here.

  • Previous studies are described in the third and fourth paragraphs of the Introduction.

5- Where are the proposed work algorithms? It should be explained in the form of an algorithm or flowchart

  • Please see the new version of Figure 2.

6-The number of references is relatively small

  • We have added to the Introduction four citations addressing related work, and an additional citation that enhances one of these extra citations.

7-The topic is modern in terms of its presentation. Still, it was not presented clearly, and the practical approach was not clarified in the accepted format when working on building an automated learning system. However, it requires a little organization and clarification for each detail of the work to be perfectly ready for publication. Likewise, the work did not address any previous works, but this matter is important because any work must be preceded by previous works that gave him an idea to build future work.

  • Previous studies are described in the third and fourth paragraphs of the Introduction. We also improved several other areas of the paper.

Reviewer 2 Report

Comments and Suggestions for Authors

Under-coding or unlabeling is indeed an important issue in EHR data. This article develops a hybrid NLP tool to detect problematic opioid use and examine the population characteristics of patients identified by this tool against those identified by ICD codes. It is quite interesting to read this paper. I only have a few comments below.

1. Please clearly explain the reasoning of how you classify patients as "negative" of problematic opioid use. If my understanding is correct, a patient who is prescribed opioids or has opioids on the current medication list, is considered negative for problematic opioid use disorder in your study - you believe that the patient is using opioids properly. The one-sentence definition in L103-104 of P3 does not intuitively explain this. Note that this assumption is also debatable since many OUD patients may still get prescriptions for opioids.   

2. Have you considered creating key phrases of medications for OUD (e.g., buprenorphine, methadone, naltrexone) for your rule-based tool? A patient on any MOUD is probably positive for problematic opioid use. 

3. Please enrich the limitations or future work subsections. You have a large dataset but your hybrid NLP tool is actually trained and evaluated on a small subset (with hundreds of snippets). It is possible that your tool does not perform well when you applied it on the remaining and much more snippets in the dataset. There has been significant progress in recent years on large language models, transfer learning, and semi-supervised learning, etc. These techniques are not used in this study. It is better to have some discussions on these as well.

Author Response

We thank you for your thoughtful review of the manuscript.  Below are responses to your comments.

  1. Please clearly explain the reasoning of how you classify patients as "negative" of problematic opioid use. If my understanding is correct, a patient who is prescribed opioids or has opioids on the current medication list, is considered negative for problematic opioid use disorder in your study - you believe that the patient is using opioids properly. The one-sentence definition in L103-104 of P3 does not intuitively explain this. Note that this assumption is also debatable since many OUD patients may still get prescriptions for opioids. 
  • You are correct in your understanding. We did not consider any patient receiving opioids as having problematic use unless there was additional documentation for it.  We now include additional clarification as to what type of text was considered positive for problematic opioid use, namely that this positive text is generally consistent with clinical or standard language that is readily recognized.  We thought that the best way to further communicate this is through examples, which are now included in the relevant paragraph, as well as Table 3.  Also, you are correct – OUD patients may still get prescriptions for opioids – we took this into account as we analyzed snippets.  
  1. Have you considered creating key phrases of medications for OUD (e.g., buprenorphine, methadone, naltrexone) for your rule-based tool? A patient on any MOUD is probably positive for problematic opioid use.
  • Our initial list of 46 key phrases included many additional drug names. The final list of key phrases were the most relevant for this data.  However, your suggestion is well-taken and is always good advice; in future research in this domain we will again include many expressions of opioid drugs and their misuse in the beginning analytical stages of our work.
  1. Please enrich the limitations or future work subsections. You have a large dataset but your hybrid NLP tool is actually trained and evaluated on a small subset (with hundreds of snippets). It is possible that your tool does not perform well when you applied it on the remaining and much more snippets in the dataset. There has been significant progress in recent years on large language models, transfer learning, and semi-supervised learning, etc. These techniques are not used in this study. It is better to have some discussions on these as well.
  • These are all very good points. We augmented the Limitations section to address the test set size.  Although it was small in comparison to the amount of notes the tool classified, it was evident in assessing the output that the NLP tool performed well. In future work we hope to revisit this study, experimenting with other machine learning models, including deep learning, and large language models if this can be facilitated in the secured environment in which we do this research.  We now mention this in the Future Work section.

Reviewer 3 Report

Comments and Suggestions for Authors

1. Most of the references used in the discussion related to Opioid use disorder are old. Authors need to include new papers related to this area and present the findings. 

2. Discussion section need to be strengthened. It does not highlight critical findings and research gaps related to Opioid use disorder. 

3. NLP pipeline is very abtract. Authors need to clearly explain the intermittent steps, detailed analysis on the different classifiers/algorithms as part of NLP track. 

4. Preprocessing details of raw text need to be included. 

5. Performance analysis with respect to SOTA models needs to be included. 

Comments on the Quality of English Language

Need to be improved. 

Author Response

We thank you for your thoughtful review of the manuscript.  Below are responses to your comments.

  1. Most of the references used in the discussion related to Opioid use disorder are old. Authors need to include new papers related to this area and present the findings.
  • After searching PubMed again we identified four studies relevant to our work that we have added – three to the Introduction, and one to the Discussion. Thank you for this suggestion. To improve readability we separated the content in the Introduction describing others’ related research into two paragraphs.
  1. Discussion section need to be strengthened. It does not highlight critical findings and research gaps related to Opioid use disorder.
  • The gap that we noticed in our research, how problematic opioid use documentation affects outcomes, inspired this study. In the Discussion we also include findings from another study we recently discovered (thanks to your previous suggestion) that also addresses this gap.
  1. NLP pipeline is very abstract. Authors need to clearly explain the intermittent steps, detailed analysis on the different classifiers/algorithms as part of NLP track. 
  • We updated Figure 2 and its legend, so now they provide this information in more detail.
  1. Preprocessing details of raw text need to be included.
  • After the rule-based module classifies snippets, the remaining unclassified snippets are preprocessed for the trained support vector machine model. The details on this, including what was done to preprocess the text prior to machine learning, is in the first paragraph under 2.3 NLP Tool Development, and further explanation on when the preprocessing occurs is noted on lines 180 – 181 on page six. We also improved Figure 2 so it now graphically indicates where preprocessing occurs.
  1. Performance analysis with respect to SOTA models needs to be included.
  • We now explicitly do this in the Discussion (please see paragraph seven).

Reviewer 4 Report

Comments and Suggestions for Authors

First and foremost, I appreciate the innovative approach taken in this study, utilizing NLP to detect opioid use. However, I must highlight several limitations in the research design.

Firstly, although the authors effectively captured symptoms of opioid abuse using NLP, the comparative analysis presented may not be sufficiently rigorous for an academic journal.

Secondly, the statistical model primarily describes differences in demographic characteristics and clinical comorbidities. I believe these aspects should be combined to support the authors' hypothesis.

I would like to suggest that the authors focus more on utilizing NLP to identify demographic and socioeconomic characteristics associated with opioid abuse, rather than relying solely on t-tests or chi-square tests.

Author Response

We thank you for your thoughtful review of the manuscript.  Below are responses to your comments.

Firstly, although the authors effectively captured symptoms of opioid abuse using NLP, the comparative analysis presented may not be sufficiently rigorous for an academic journal.

  • We followed procedures that we know of that are standard to the domain to maintain academic rigor. We followed standard practice in text annotation, including measuring annotator inter-rater agreement and iterating the process until high agreement was reached. We followed standard practices in NLP tool development, data collection and processing.  (We improved Figure 2 to better describe the NLP pipeline). As for the comparison between the patients identified only through NLP, patients receiving a relevant ICD code, and patients having no problematic opioid use documentation, we used standard chi-square tests and t tests. To add rigor (because of the large sample sizes), we also measured the absolute standardized difference (ASD).  We hope that the modifications we made to this revision clarify what we did.

Secondly, the statistical model primarily describes differences in demographic characteristics and clinical comorbidities. I believe these aspects should be combined to support the authors' hypothesis.

  • Thank you for this suggestion. We modified the first paragraph in the Discussion to clearly indicate this.

I would like to suggest that the authors focus more on utilizing NLP to identify demographic and socioeconomic characteristics associated with opioid abuse, rather than relying solely on t-tests or chi-square tests.

  • This is a great suggestion. In fact, we are interested in incorporating social determinants of health into future studies.  We now mention this in Future Work.

Round 2

Reviewer 1 Report

Comments and Suggestions for Authors

All comments are done.

Reviewer 3 Report

Comments and Suggestions for Authors

Suggestions are addressed 

Comments on the Quality of English Language

Minor